# OpenReview forum: "AutoTool: Dynamic Tool Selection and Integration for Agentic Reasoning"
_ICLR.cc/2026/Conference — Submitted to ICLR 2026_

### Official Review · Reviewer_Rg4C · 2025-10-30

**Soundness:** 2
**Presentation:** 3
**Contribution:** 2
**Rating:** 4
**Confidence:** 4

**Summary:**

This paper introduces AutoTool, a method designed to evoke dynamic tool selection and integration in LLM reasoning, aiming for development of robust LLM agents under evolving toolsets. AutoTool constructs a 200k agentic reasoning dataset that contains a wide collection of tools and tasks, with tool-selection rationales. Based on that, it develops a dual-phase training scheme for LLM agents, including a SFT + RL phase to stabilize the learning of tool-integrated reasoning trajectories, followed by a KL-regularized Plackett–Luce (PL) ranking phase to refine the tool-selection part of the reasoning. AutoTool achieves performance gains on two LLMs (Qwen3-8B and Qwen2.5-VL-7B) in a diverse set of reasoning tasks, outperforming other advanced
LLM agents and baseline tool-integration methods.

**Strengths:**

- AutoTool innovatively integrates embedding-anchored tool selection and KL-regularized PL ranking into the learning of LLM agents, which contributes to decent originality.
- The presentation of AutoTool dual-phase learning scheme is theoretically well-motivated and mathematically well-grounded.
- AutoTool’s proposed challenge of dynamic tool selection under evolving tool environments is crucial for robust and scalable LLM agentic framework development.

**Weaknesses:**

- The experimental analysis of this paper falls short of justifying AutoTool’s effectiveness on improving dynamic tool selection under evolving tool environments, i.e., whether AutoTool performs better tool selection when generalizing to unseen toolsets, which is however the most significant challenge raised by the paper. Evaluation on a new or heldout set of tools and tasks that are unseen at training phase would help further justify this important point.
- It is unclear how the evolving toolset T with dynamic size is constructed for AutoTool learning, i.e., for each training sample or question, which candidate tools are chosen to form the evolving toolset and how to control a decent tool-selection difficulty regarding to the number or proportion of useful and irrelevant tools in the toolset. The design of evolving toolset T at training phase is crucial for learning the dynamic tool selection.
- There is no quantitative analysis to verify the positive correlation between the tool selection accuracy and the final answer accuracy. It would be better to more directly justify that, compared to baseline methods, AutoTool has a better hit rate of selecting the correct oracle tool, and this contributes to its better final performances.

**Questions:**

- Any additional experimental results to resolve the above weaknesses?
- Is there an ablation study to measure how many performance gains are due to the incorporation of additional tool-selection rationales introduced in AutoTool?
- Are there any qualitative or quantitative comparisons with regard to the scope of toolsets studied in AutoTool and in other related work of tool integration, such as ToolLLM, RestGPT and HuggingGPT?

---

> ### Author Response · Authors · 2025-11-21
> **Author Rebuttal (Part 1/4)**
>
> **Dear Reviewer Rg4C,**
>
> Thank you very much for your detailed feedback and for acknowledging the strengths of our work! We have thoroughly revised the paper in light of your comments and added several complementary experiments and analyses.  **In the updated paper, all modifications are marked in blue for easy reference.**
>
> Below, we provide point-by-point responses to address each of your questions and concerns. **For better clarity, we also include a concise Roadmap before the extended explanations**.
>
> ---
> ## **W.1 Generalization of AutoTool on Unseen Tool Environments**
> ## Roadmap
> We thank the reviewer for the insightful question! We would like to clarify that generalizing AutoTool to evolving unseen tools is indeed a central goal of our work, and we have incorporated this setting throughout our current experiments. As a response to address your question:
>
> - **First,** we clarify how our data curation pipeline separates toolset construction from training instance collection, ensuring that **65.8% of the tools used at inference are entirely unseen during training**.
> - **Second,** we elaborate how **our current experiments demonstrate that AutoTool actively selects unseen tools during downstream inference.**
> - **Third,** we further demonstrate AutoTool's strong tool-selection ability by **conducting new experiments in which all inference-time tools are completely unseen during training**.
>
> ## Response
>
> **(i) Our Current Data Curation Setups to Ensure Evaluation on Unseen Tools**
>
> In `Section 2 (Data Curation for Tool-Selection)`, we separate our data curation process into three stages.
>
> ||Number of Tools|
> |-|-|
> |Full Curated ToolSet|1346|
> |Tools used for AutoTool dual-stage training|460 (34.2%)|
> |**Unseen Tools used only at inference**|**886 (65.8%)**|
>
> - For **Stage 1: Toolset & Task Collection**, we collect **1346 tools** from diverse sources (online APIs, open-source libraries, etc.). Notably, the entire toolset is not used during training—only a small subset of **460 predefined tools** are seen during training, while the remaining tools are unseen.
> - For **Stages 2 & 3: Tool-Selection Rationale & Augmentation**, we use existing tool-integration datasets that already contain the **460 predefined tools**. These datasets provide high-quality CoT traces paired with these predefined tools, and we simply augment each instance with tool-selection rationales. **Notably, only this small subset of 460 tools with the complete CoT instances is used for AutoTool training,** while our full toolset contains 1346 tools, and **the rest 65.8% of unseen tools are never used during training.**
>
>
> **Toolset Selection During Training & Inference:**\
> During training, we use only the **460 predefined tools** as the selection pool. For downstream inference, we expand the selection pool to the **full toolset of 1346 tools** to ensure broad coverage across all downstream tasks. **This constitutes the evolving, unseen toolset from which AutoTool will select tools during inference.**
>
>
> **(ii) Interpretation of Current Experiments to Highlight Generalization on Unseen Tools**
>
> To better interpret the existing experiments in `Table 1 (Section 4)`, we **conduct an additional analysis to investigate how well AutoTool can dynamically adapt to select unseen tools within the full toolset**.
>
> **[Experiment Setup]**
>
> We analyze AutoTool's recorded trajectories from our previous experiments by extracting tool-selection steps and identifying the selected tools. Using these, we **measure the proportion of unseen versus seen tools that lead to correct final answers across the 7 downstream tasks**, using AutoTool (Qwen3-8B) under the same experimental settings as in our paper.
>
> **[Evaluation Result]**
>
> | Propotion of Selected Unseen/Seen Tools (%)|AIME24|AIME25|HotpotQA|2Wiki|V-Chart|V-Math|V-Code|
> |-|-|-|-|-|-|-|-|
> |AutoTool selects Seen Tools|**57.4**|41.8|27.6|18.1|14.5|46.8|26.5|
> |**AutoTool selects Unseen Tools**|42.6|**58.2**|**72.4**|**81.9**|**85.5**|**53.2**|**73.5**|
>
> **Key Findings:**
> - From the table, we observe that **AutoTool actively leverages a higher proportion of unseen tools on downstream tasks**, especially for search-heavy and multimodal understanding tasks.
> - When combined with `Table 1`, these results show that AutoTool’s strong downstream performance **is largely driven by its ability to correctly select unseen tools**, rather than solely memorizing the tools observed during training.
>
> ***In our next response to the reviewer, we continue on this question and further deploy AutoTool on new datasets with fully unseen toolsets.***

---

> ### Author Response · Authors · 2025-11-21
> **Author Rebuttal (Part 2/4)**
>
> ## **W.1 (Continuous)**
> **(iii) New Experiments with Fully Unseen Toolsets Setting**
>
> We further conduct new experiments where **all available tools at inference time are completely unseen during training** to comprehensively demonstrate AutoTool’s generalizability.
>
> **[Experiment Setup]**\
> To simulate a fully unseen-tool setting, we reuse the same downstream benchmarks (AIME24, V-Chart) but **restrict the inference-time tool pool to the subset of 886 tools that are never used during training**. This ensures that all tools utilized by AutoTool during inference are fully unseen.
>
> We further evaluate AutoTool on **two new out-of-domain datasets LiveCodeBench-V2 [1] and MedQA [2], with complete unseen tools** and compare with the strong baselines in our paper. We use AutoTool (Qwen3-8B) as our evaluation model and keep all experiment settings the same as in our paper.
>
> **[Evaluation Result]**
>
> |Method|AIME24|HotpotQA|LiveCodeBench|MedQA|
> |-|-|-|-|-|
> |Search-R1|13.3|43.3|36.2|73.5|
> |ReTool|66.7|21.4|71.5|64.7|
> |**AutoTool (with unseen tools)**|**66.7**|**43.8**|**77.9**|**84.3**|
>
>
> - From the table, **AutoTool under the unseen-tools setting still achieves the strongest performance across all task domains (including the two new datasets)**, when compared with baseline tool-integration methods.
> - When compared with AutoTool using the full toolset (1346 tools), we observe only **a marginal performance drop** of less than 2% on both AIME24 and HotpotQA.
>
>
> **Summary.** Our explanation of toolset curation, tool-usage analyses, and new unseen-tool experiments demonstrate that AutoTool generalizes its tool selection to evolving unseen toolsets.
>
> *Kind Note: We have revised our writing in `Section 2` ([click here](https://openreview.net/pdf?id=52c4trAbmd#page=2.81)) and `Appendix F`([click here](https://openreview.net/pdf?id=52c4trAbmd#page=24.10)) to clearly explain our seen-unseen tool split and evolving toolset construction.*
>
> **References:**
>
> [1] Livecodebench: Holistic and contamination free evaluation of large language models for code.\
> [2] Llm-medqa: Enhancing medical question answering through case studies in large language models.
>
> ---
> ## **W.2 Clarifying the Construction of the Evolving Toolset**
> ## Roadmap
> Thank you for the insightful question on our evolving toolset. Below, we clarify (i) how the toolset is constructed for each training sample and during inference, and (ii) how we control tool-selection difficulty in a principled and reproducible manner.
>
> ## Response
> **(i) How the Evolving Toolset Is Constructed**
> > *“For each training sample or question, which candidate tools are chosen to form the evolving toolset \(T\)?”*
>
> *Note: In our early response to `Reviewer Rg4C W.1`, we have clarified that the full toolset contains 460 seen tools and 886 unseen tools.*
>
> **During training**, the toolset is *not* evolving. It remains fixed to the **460 seen tools**, because all tool-selection rationales and ground-truth tool trajectories come from source datasets that include only these 460 predefined tools. Thus, **every training sample uses the same 460-tool candidate pool**, which ensures **stable tool-embedding learning and consistent PL-ranking optimization**.
>
> Importantly, AutoTool learns **dynamic tool-selection behavior through its embedding-based matching mechanism**, not through changes in the training-time toolset. Therefore, an evolving toolset is **not required** for learning dynamic selection.
>
> **During inference**, AutoTool does not assume a fixed or closed tool inventory. In realistic deployments, the available tools may change as new APIs are added or domain registries are updated. To evaluate tool-selection capability in a controlled and reproducible manner, we instantiate the evolving-toolset assumption using the original 460 seen tools together with a held-out pool of 886 unseen tools. Although this unseen pool is fixed for benchmarking, it serves as a **principled proxy** for the appearance of new tools at inference time, allowing us to simulate an evolving tool environment while ensuring fairness across all baselines.
>
>
> **(ii) How We Control Tool-Selection Difficulty**
> > *“How to control a decent tool-selection difficulty regarding the number or proportion of useful and irrelevant tools?”*
>
> We control tool-selection difficulty by adjusting:
> - the **size** of the candidate toolset,
> - the **ratio** of task-relevant vs. irrelevant tools,
> - the **semantic similarity** among tools, and
> - the **degree of expansion** from training to inference.
>
> In our experimental setting, **the tool-selection difficulty is intentionally high**: among the 1346 tools available at inference, **only a small subset is task-relevant for any given question**, while the majority belong to domains unrelated to that specific downstream query. This results in a challenging and realistic selection environment for AutoTool and all baselines.

---

> ### Author Response · Authors · 2025-11-21
> **Author Rebuttal (Part 3/4)**
>
> ## **W.3 Analyze Correlation between Tool Selection and Final Accuracy**
> ## Roadmap
> Thank you for the insightful suggestion. Following the reviewer’s recommendation, we conduct **an additional analysis examining the relationship between the intermediate tool-selection hit rate and the LLM agent’s final performance**. We detail our experiments and findings below:
>
> ## Response
> **[Experiment Setup]**\
> Following our experiment setup in `Table 3`, we perform the analyses on 5 downstream datasets covering different tasks, including AIME24, HotpotQA, 2Wiki, MMSearch, and V-Chart. We use AutoTool (Qwen3-8B) as our evaluation model and keep all experiment settings the same as in our paper.
>
> For each testing question, we first **collect the entire output trajectories** generated by AutoTool and three other Tool-integration baselines (Search-R1, v-ToolRL, and ReTool). Then for each method, **we extract the tool-selection step and compare with the oracle tool for the hit rate evaluation**.
>
> We define the **Hit Rate** metric for AutoTool as:
> $
> \text{HitRate} = \frac{
> \text{Num of }(\text{steps where } t\_i^{\text{AutoTool}} = t\_i^{\text{oracle}})
> }{
> \text{Num of } (\text{all tool-selection steps})
> }
> $
>
> **[Evaluation Results]**
> |HitRate(%)|AIME24|HotpotQA|2Wiki|MMSearch|
> |-|-|-|-|-|
> |Search-R1|10.1|88.7|81.2|-|
> |v-ToolRL|-|-|-|64.3|
> |ReTool|62.5|56.2|47.8|-|
> |AutoTool (Phase I only)|60.3|81.5|77.6|71.4|
> |**AutoTool (Phase I + II)**|**77.4**|**91.6**|**85.8**|**76.9**|
>
> *Note: "-" denotes the method cannot be applied to the specific task.*
>
> When comparing the new analysis results with the final accuracies reported in `Table 1` and `Table 2` in our paper, we observe two consistent trends:
> - **Across baselines**, there is a **clear positive correlation between tool-selection hit rate and downstream performance**. Tool-integration methods that fail to generalize across tasks (e.g., Search-R1 on AIME24 or ReTool on HotpotQA) exhibit low hit rates on their unfamiliar tasks, which correspondingly cause low final accuracies, as shown `Table 1`.
> - **Within AutoTool**, comparing *Phase I only* vs. *Phase I + II* shows that **Phase-II PL-ranking refinement greatly improves tool-selection hit rate, which directly leads to higher end-task performance** as reported in `Table 2`.
>
> Our two-dimensional comparison demonstrates that AutoTool’s improvements in intermediate tool-selection accuracy consistently translate into higher final answer quality.
>
> ---
> ## **Q.1 Address Weakness Above**
> Thank you for the question! We have presented additional experiment results and detailed analyses to each of the listed questions above.
>
> ---
> ## **Q.2 Ablation on Tool Selection Rationales**
> ## Roadmap
> We thank the reviewer for asking about the effectiveness of the tool-selection rationales in AutoTool. We provide a detailed ablation study below.
> ## Response
> **[Experiment Setups]**\
> To isolate the contribution of the tool-selection rationales, we **compare the full AutoTool system against a version without rationales**, while keeping all other modules the same. We consider the following variants and evaluate across AIME24, HotpotQA, and V-Chart:
> - **AutoTool:** We follow our original method to ask the model (i) first generates an explicit tool-selection rationale to justify which tool among the candidates should be used, and then (ii) predicts the corresponding tool-name anchor token.
> - **AutoTool (w/o rationale):** We **remove the entire tool-selection rationale from the training data**. The model is trained to **directly predict the tool name** without generating any intermediate reasoning. All other components, such as the model architecture, tool descriptions, and tool specifications, remain exactly the same.
>
> **[Evaluation Results]**
>
> | Method (Qwen3-8B)| AIME24|GPQA-Diamond |HotpotQA | V-Chart|
> |-|-|-|-|-|
> |AutoTool (w/o rationale)|60.0|71.2|41.8|22.6|
> |**AutoTool**|**66.7**|**73.7**|**45.1**|**24.7**|
>
> From the experiment, we observe that:
> - Across all four benchmarks, adding tool-selection rationales consistently improves model performance. The gains are strongest on reasoning-intensive tasks such as AIME24 and HotpotQA, indicating that **explicit justification helps the model choose the correct tool in more complex scenarios.**
> - We further analyze all tool-invocation cases and find that adding the rationale increases successful tool executions by 13.5%. This indicates that **incorporating selection rationales leads to more reliable tool use during intermediate steps.**
>
> Overall, these results demonstrate that the tool-selection rationale is **a key component** for achieving more accurate and robust tool use in AutoTool.

---

> ### Author Response · Authors · 2025-11-21
> **Author Rebuttal (Part 4/4)**
>
> ## **Q.3 ToolSets Comparison with Prior Works**
> ## Roadmap
> Thanks for your careful reading of our toolsets construction and the suggestions on related work comparison! Below, we provide a detailed analysis of AutoTool and other toolsets from ToolLLM [1], RestGPT [2], and HuggingGPT [3].
> ## Response
> |Toolset|# of Tools/tasks|Support Training towards Tool Selection|Dynamic Tool Environment| High-quality Tool-Selection Rationales|
> |-|-|-|-|-|
> | ToolLLM [1] | 3,451 tools (API only), 49 tasks | can only select online APIs|No|No|
> | RestGPT [2]| 100+ tools (API only), 2 tasks |No|No|No|
> | HuggingGPT [3]| 400+ tools, 24 tasks |No|No|No|
> | **AutoTool (ours)** | **1,000+ tools** across **100+ tasks**|**Yes (both online and offline tools)** |**Yes** | **Yes**|
>
> Based on the comparison, we highlight two key points of AutoTool:
> - AutoTool integrates a relatively **large-scale collection of 1,000+ multimodal tools**, surpassing the scope of RestGPT (~100) and HuggingGPT (400+). While ToolLLM contains more tools, AutoTool covers the **widest range of tasks** and incorporates **both online and offline tools**. Crucially, unlike all compared works, our dataset includes **high-quality tool-selection rationales**, providing explicit supervision on why a specific tool is chosen.
> - More importantly, our toolset uniquely supports training towards **proactive tool selection**. Unlike ToolLLM (which relies on static data accessed via a retriever) or RestGPT/HuggingGPT (which are training-free), AutoTool’s toolset is specifically built upon a **Dynamic Tool Environment**. This data structure inherently supports **generalization to novel, unseen tools** that were not present in the initial pool.
>
> *Kind Note: We follow the reviewer’s suggestion and have integrated the above discussions, providing a more thorough comparison with prior works in our revised paper. (`Appendix G` [click here](https://openreview.net/pdf?id=52c4trAbmd#page=25.10))*
>
> **References:**\
> [1] ToolLLM: Facilitating Large Language Models to Master 16000+ Real-world APIs. \
> [2] RestGPT: Connecting Large Language Models with Real-World RESTful APIs.\
> [3] HuggingGPT: Solving AI Tasks with ChatGPT and its Friends in HuggingFace.
>
> ---
> ## Happy to have further discussion!
> **Thank you again for the thoughtful review to help us enhance the paper quality. We hope our responses address your concerns, and we are happy to discuss if you have any further questions!**

---

> ### Comment · Reviewer_Rg4C · 2025-11-24
>
> Thank you for your response which addresses or clarifies my concerns. I have updated my rating.

---

> > ### Author Response · Authors · 2025-11-24
> > **Thanks for the positive feedback!**
> >
> > Dear Reviewer Rg4C,
> >
> > Thank you very much for the positive rating and thoughtful follow-up feedback! We are glad that our clarifications fully addressed your questions, and we sincerely appreciate your time, effort, and support of our work!
> >
> > Warm regards,
> >
> > The Authors of AutoTool

---

### Official Review · Reviewer_HQaG · 2025-10-31

**Soundness:** 3
**Presentation:** 3
**Contribution:** 3
**Rating:** 4
**Confidence:** 4

**Summary:**

This paper introduces a dynamic selection method for tools in agentic large language models. Typical agentic models assume a fixed set of tools to use in their reasoning process. This paper introduces a dynamic selection process where the model can utilize a large set of tools through retrieval. The proposed approach can also handle new tools which can be added to the tool repository for the models to use. Empirical results show the dynamic tool selection method outperforms existing tool-integration methods and can generalize well to unseen tools during inference.

**Strengths:**

- Comprehensive empirical results, spanning a diverse set of evaluation datasets
- Results compared against relevant baselines such as stronger reasoning models, existing tool integration methods and traditional fine-tuning
- Strong results, the proposed AutoTool framework achieves consistent gains on the diverse datasets compared to multiple approaches.

**Weaknesses:**

- I couldn't find the results on the generalization performance on unseen tools during inference. The key proposal for the embedding-anchored selection method is that it should be able to dynamically adapt to new tools provided during inference, but none of the experimental results seem to highlight it.
- Not sure I follow why the analysis of autotool is needed with an oracle tool assignment agent. Ideally, the oracle numbers should be present in Table 1 to directly compare other methods on how close they too are with the oracle assignment, if its necessary.

While overall the paper and contribution is good, its missing this key ingredient (generalization) - I'm ready to raise my scores if it is presented and analyzed comprehensively.

**Questions:**

Same as my weakness - where is the generalization result? I think that should be the key result to highlight, along with analysis where the generalization works and fails.

---

> ### Author Response · Authors · 2025-11-21
> **Author Rebuttal (Part 1/2)**
>
> **Dear Reviewer HQaG,**
>
> Thank you very much for your insightful feedback and for acknowledging the strengths of our work! We have thoroughly revised the paper in light of your comments and added several complementary experiments and analyses.  **In the updated paper, all modifications are marked in blue for easy reference.**
>
> Below, we provide detailed, point-by-point responses addressing each of your questions and concerns. **For more convenient reading, we also include concise Roadmap summaries before the extended explanations whenever appropriate.**
>
> ---
> ## **W.1 & Q.1 AutoTool's Generalization on Unseen Tools**
> ## Roadmap
> We thank the reviewer for the insightful question on AutoTool’s generalization ability! We would like to clarify that generalizing AutoTool to unseen tools is indeed a central goal of our work, and we have incorporated this setting throughout our current experiments. As a response to address your question:
>
> - **First,** we clarify how our data curation pipeline separates toolset construction from training instance collection, ensuring that **65.8% of the tools used at inference are entirely unseen during training**.
> - **Second,** we elaborate how **our current experiments demonstrate that AutoTool actively selects unseen tools during downstream inference.**
> - **Third,** we conduct **new experiments where all inference-time tools are completely unseen during training** to further evaluate AutoTool’s generalization ability.
>
> ## Response
>
> **(i) Our Current Data Curation Setups to Ensure Evaluation on Unseen Tools**
>
> In `Section 2`, we separate our data curation process in three stages.
>
> ||Number of Tools|
> |-|-|
> |Full Curated ToolSet|1346|
> |Tools used for AutoTool dual-stage training|460 (34.2%)|
> |**Unseen Tools used only at inference**|**886 (65.8%)**|
>
> - For **Stage 1: Toolset & Task Collection**, we collect **1346 tools** from diverse sources (online APIs, open-source libraries, etc.). Notably, the entire toolset is not used during training—only a small subset of **460 predefined tools** are seen during training, while the remaining tools are unseen.
> - For **Stages 2 & 3: Tool-Selection Rationale & Augmentation**, we use existing tool-integration datasets that already contain the **460 predefined tools**. These datasets provide high-quality CoT traces paired with these predefined tools, and we simply augment each instance with tool-selection rationales. **Notably, only this small subset of 460 tools with the complete CoT instances is used for AutoTool training,** while our full toolset contains 1346 tools, and **the rest 65.8% of unseen tools are never used during training.**
>
>
> **Toolset Selection During Training & Inference:**\
> During training, we use only the **460 predefined tools** as the selection pool. For downstream inference, we expand the selection pool to the **full toolset of 1346 tools** to ensure broad coverage across all downstream tasks. **This constitutes the evolving, unseen toolset from which AutoTool will select tools during inference.**
>
> ***In our next response to the reviewer, we continue on this question and further demonstrate that AutoTool indeed actively leverages more unseen tools than seen ones during inference.***

---

> ### Author Response · Authors · 2025-11-21
> **Author Rebuttal (Part 2/2)**
>
> ## **W.1 & Q.1 (Continuous)**
>
> **(ii) Interpretation of Current Experiments to Highlight Generalization on Unseen Tools**
>
> To better interpret the existing experiments in `Table 1 (Section 4)`, we **conduct an additional analysis to investigate how well AutoTool can dynamically adapt to unseen tools within the full toolset**.
>
> **[Experiment Setup]**
>
> We analyze AutoTool's recorded trajectories from our previous experiments by extracting tool-selection steps and identifying the selected tools. Using these, we **measure the proportion of unseen versus seen tools that lead to correct final answers across the 7 downstream tasks**, using AutoTool (Qwen3-8B) under the same experimental settings as in our paper.
>
> **[Evaluation Result]**
>
> | Propotion of Selected Unseen/Seen Tools (%)|AIME24|AIME25|HotpotQA|2Wiki|V-Chart|V-Math|V-Code|
> |-|-|-|-|-|-|-|-|
> |AutoTool selects Seen Tools|**57.4**|41.8|27.6|18.1|14.5|46.8|26.5|
> |**AutoTool selects Unseen Tools**|42.6|**58.2**|**72.4**|**81.9**|**85.5**|**53.2**|**73.5**|
>
> **Key Findings:**
> - From the table, we observe that **AutoTool actively leverages a higher proportion of unseen tools on downstream tasks**, especially for search-heavy and multimodal understanding tasks.
> - When combined with `Table 1`, these results show that AutoTool’s strong downstream performance **is largely driven by its ability to correctly select unseen tools**, rather than simply memorizing the tools observed during training.
>
> **(iii) New Experiments with Fully Unseen Toolsets Setting**
>
> We further conduct new experiments where **all available tools at inference time are completely unseen during training** to comprehensively demonstrate AutoTool’s generalizability.
>
> **[Experiment Setup]**\
> To simulate a fully unseen-tool setting, we reuse the same downstream benchmarks (AIME24, V-Chart) but **restrict the inference-time tool pool to the subset of 886 tools that are never used during training**. This ensures that all tools utilized by AutoTool during inference are fully unseen.
>
> We further evaluate AutoTool on **two new out-of-domain datasets LiveCodeBench-V2 [1] and MedQA [2], with 886 complete unseen tools** and compare with the strong baselines in our paper. We use AutoTool (Qwen3-8B) as our evaluation model and keep all experiment settings the same as in our paper.
>
> **[Evaluation Result]**
>
> |Method|AIME24|HotpotQA|LiveCodeBench|MedQA|
> |-|-|-|-|-|
> |Search-R1|13.3|43.3|36.2|73.5|
> |ReTool|66.7|21.4|71.5|64.7|
> |**AutoTool (with unseen tools)**|**66.7**|**43.8**|**77.9**|**84.3**|
>
> - From the table, **AutoTool under the unseen-tools setting still achieves the strongest performance across all task domains (including the two new datasets)**, when compared with baseline tool-integration methods.
> - When compared with AutoTool using the full toolset (1346 tools), we observe only **a marginal performance drop** of less than 2% on both AIME24 and HotpotQA.
>
> **Summary.**  Our explanation of toolset curation, tool-usage analyses, and new unseen-tool experiments demonstrate that AutoTool generalizes its tool selection to evolving unseen toolsets.
>
> *Kind Note: We have revised our writing in `Section 2` ([click here](https://openreview.net/pdf?id=52c4trAbmd#page=2.81)) and `Appendix F`([click here](https://openreview.net/pdf?id=52c4trAbmd#page=24.10)) to clearly explain our seen-unseen tool split and evolving toolset construction.*
>
> **References:**
>
> [1] Livecodebench: Holistic and contamination free evaluation of large language models for code.\
> [2] Llm-medqa: Enhancing medical question answering through case studies in large language models.
>
> ---
> ## **W.2 Explanation on Oracle Tools Comparison**
> ## Roadmap
> We explain our original motivation for separately analyzing the oracle comparison and follow the reviewer’s suggestion by incorporating the oracle results directly into`Table 1`.
> ## Response
> Thank you for the insightful question regarding the role of the oracle tool-assignment comparison.
> - **To clarify our original intention in the paper:** we reported the oracle results separately because (i) such direct comparison between AutoTool and the oracle setting is straightforward to highlight AutoTool’s dynamic tool-selection ability, showing that it reaches performance close to the oracle optimal upper bound, and (ii) several baselines without tool-selection capabilities might not be properly paired with oracle-provided tools, since their methods do not support utilizing tools that were unseen during training.
> - That said, we fully agree that including oracle numbers in Table 1 improves clarity. **In the revised paper, we have incorporated the oracle results directly into the main table**, enabling a transparent side-by-side comparison and making the performance gap between each method and an oracle assignment immediately visible.
>
> ---
> ## Happy to have further discussion!
> **Thank you again for the thoughtful review. We hope our responses address your concerns and are happy to discuss any further questions!**

---

> ### Author Response · Authors · 2025-11-27
> **Looking Forward to Your Insightful Feedback!**
>
> Dear Reviewer HQaG,
>
> We sincerely thank you for your thoughtful and constructive review, which has greatly helped us improve our manuscript! In response to your concerns, we have conducted additional experiments and provided in-depth analyses of our method.
>
> As the discussion period draws to a close, we would appreciate your feedback on whether our responses have addressed your concerns. We are fully committed to addressing any remaining questions you may have. Thank you very much for your time and consideration of our research!
>
> Warm regards,
>
> Authors of AutoTool

---

### Official Review · Reviewer_NAkE · 2025-11-01

**Soundness:** 3
**Presentation:** 2
**Contribution:** 3
**Rating:** 6
**Confidence:** 3

**Summary:**

This paper presents AutoTool, a framework that equips LLM agents with dynamic tool-selection capabilities throughout their reasoning trajectories. The authors construct a 200k dataset with explicit tool-selection rationales across 1,000+ tools and 100+ tasks, then employ a dual-phase optimization pipeline: (i) trajectory stabilization via SFT and RL, and (ii) KL-regularized Plackett-Luce ranking for tool selection refinement. Experiments show consistent improvements across math, science, code generation, and multimodal benchmarks.

**Strengths:**

The paper addresses a genuine limitation in existing work—most approaches assume fixed toolsets, whereas real-world scenarios require dynamic tool selection from evolving inventories.

The dual-phase optimization pipeline is well-designed, with Phase I establishing stable reasoning patterns and Phase II specifically targeting tool-selection refinement through PL ranking.

**Weaknesses:**

While the combination is effective, the individual components (SFT, GRPO, Plackett-Luce ranking) are well-established techniques. The main contribution appears to be applying PL ranking to tool selection, which is somewhat incremental. The paper would benefit from discussing recent work on tool retrieval and generation.
Also there are notation inconsistencies: The paper switches between τ and T for trajectories/trajectory sets.

**Questions:**

I do have some scalability concerns. How does the approach scale beyond 1,000 tools? The embedding-based selection (Eq. 4) requires computing distances to all tools at each selection step.

---

> ### Author Response · Authors · 2025-11-21
> **Author Rebuttal (Part 1/2)**
>
> **Dear Reviewer NAkE,**
>
> Thank you very much for your insightful feedback and for acknowledging the strengths of our work! We have thoroughly revised the paper in light of your comments and added several complementary experiments and analyses. **In the updated paper, all modifications are marked in blue for easy reference.**
>
> Below, we provide detailed, point-by-point responses addressing each of your questions and concerns. **For your more convenient reading, we also include concise Roadmap summaries before the extended explanations whenever appropriate.**
>
> ---
> ## **W.1a AutoTool's Contribution on Dynamic Tool-Selection**
>
> ## Roadmap
> Thank you for the thoughtful question! Below, we first (i) clarify AutoTool’s motivation for using PL-ranking in dynamic tool selection, and then (ii) highlight the novelty of our PL-Ranking based optimization framework.
>
> ## Response
>
> **(i) Why PL-Ranking Is Needed in Dynamic Tool Selection?**
>
> We agree with the reviewer that SFT, GRPO, and Plackett–Luce (PL) ranking are established techniques. Moving one step forward, AutoTool’s contribution is in **adapting PL ranking to a dynamic tool-selection setting**, where the set of candidate tools evolves at inference time and contains unseen tools during training.
>
> This dynamic setting creates two challenges that prior work does not address:
> - **Challenge 1: Tool-Set Distribution Shift.** A large gap between training-time tool trajectories and unseen test-time tools.
> - **Challenge 2: Generalizability on Evolving Unseen Toolsets.** The agent must maintain stable and well-calibrated preference modeling when new and previously unseen tools appear across different tasks.
>
>
> PL-Ranking offers **a principled formulation** for aligning demonstration trajectories with policy rollouts through a step-wise preference distribution over candidate tools, thereby supporting robust generalization to unseen tools. To our knowledge, this is **the first formulation treating tool selection as a structured ranking problem under evolving toolsets**, rather than classification or retrieval over a fixed inventory.
>
> **(ii) Novelty of Our PL-Based Optimization**
>
> As we state in `Lines 307-311` and `Eq.7`, directly optimizing the PL distribution over the $N!$ permutations of tool-selection trajectories is **computationally infeasible**. To address this, our work introduces a **non-trivial theoretical bridge** that **connects PL-Ranking with LLM policy optimization**. This connection enables a **tractable surrogate objective** that faithfully preserves the underlying preference structure defined by the PL formulation.
>
> This theoretical link is essential: it allows PL-ranking to be applied in the context of **dynamic tool selection**, where the candidate toolset changes across tasks and may include unseen tools.
>
> *Note: We thank the reviewer for the question and have revised the paper (`Section 3.4` [click here](https://openreview.net/pdf?id=52c4trAbmd#page=6.46)) to state our motivation more explicitly.*
>
> ---
> ## **W.1b Discussion on Recent Works on Tool Retrieval & Generation**
> ## Roadmap
> We distinguish AutoTool from recent works in three fundamental ways, highlighting why our **PL-Ranking based formulation** is essential for evolving toolsets.
> ## Response
> |Category|Representative Works| Core Mechanism | AutoTool's Distinction|
> |-|-|-|-|
> |Retrieval|AnyTool [1]| **External Matching**: Matches queries to tools via a separate hierarchical retriever (not jointly trained with the agent policy). | **Internal Policy Alignment**: AutoTool aligns the agent's internal representations directly with tool embeddings.|
> | Generation |CRAFT [2], ToolMaker [3], CREATOR [4] | **Code Synthesis**: Generates new tools/code when suitable tools are missing.| **Selection Efficiency**: AutoTool focuses on selecting from *existing but evolving* tool pools, avoiding multi-stage coding/rectification costs.|
> |**Dynamic Tool-Selection**| **AutoTool**| **Embedding-Anchored PL Ranking**| **End-to-end adaptation to evolving/unseen toolsets** via embedding-anchored selection and listwise PL training. |
>
> **Summary:** Compared with retrieval methods that depend on external retrievers, AutoTool learns tool selection through embedding-anchored policies, enabling generalization to unseen tools. Additionally, unlike tool-generation approaches that focus on synthesizing new tools/APIs, AutoTool focuses on the complementary real-world challenge of selecting the right tool from large and evolving toolsets.
>
> *Kind Note: We provide a detailed discussion of related works in our revised paper (`Appendix H` [click here](https://openreview.net/pdf?id=52c4trAbmd#page=26.10)).*
>
> **References:**\
> [1] AnyTool: Self-Reflective, Hierarchical Agents for Large-Scale API Calls. \
> [2] CRAFT: Customizing LLMs by Creating and Retrieving from Specialized Toolsets. \
> [3] Large Language Models as Tool Makers. \
> [4] CREATOR: Tool Creation for Disentangling Abstract and Concrete Reasoning of Large Language Models.

---

> ### Author Response · Authors · 2025-11-21
> **Author Rebuttal (Part 2/2)**
>
> ## **W.2 Notation consistency for $\tau$ and $T$**
> Thank you for the detailed review! To help the reviewer better interpret notations $\tau$ and $T$ in our paper, we restate the definition for these notations utilized in our paper:
>
> - $\tau$: As defined in `Lines 177-185`, **$\tau$ refers to a complete reasoning trajectory** generated by an LLM agent.
> - $\mathcal{T}$: As defined in `Line 296`, **$\mathcal{T}$ denotes the set of collected trajectory rollouts used during PL ranking and optimization**. Each trajectory is an element of this set, i.e., $\tau \in \mathcal{T}$.
> - $T$: As defined in `Lines 173–176`, **$T$ refers to the evolving toolset available to the LLM agent**, where each tool is indexed as $t_k \in T$.
>
> *Kind Note: We follow the reviewer's suggestion to update the notation in the revised version to avoid any accidental switching of symbols.*
>
> ---
> ## **Q.1 Scalability beyond 1,000 tools**
> ## Roadmap
> We appreciate the reviewer’s thoughtful question regarding scalability. We first clarify that our method has indeed been designed with scalability in mind from multiple perspectives, and then discuss extensions to further scale AutoTool with more tools.
> ## Response
> **(i) AutoTool already scales beyond prior RL-based tool-selection work.**
> Previous RL-based tool integration methods either train models to directly invoke a *single* specific tool [1,2] or manually assign a *small and fixed* toolset (typically ≤10 tools) [3,4]. In contrast, **AutoTool extends the tool-use setting to automatically select *unseen* tools from an evolving toolset containing over 1,000 tools**. This constitutes an order-of-magnitude increase compared to prior work and demonstrates that our system can be operational at the thousand-tool scale.
>
> **(ii) The embedding-based selection step is efficient in practice.**
>
> Although Eq. 4 computes similarity against all tools, this step operates entirely on **frozen, pre-computed tool embeddings** (as stated in `Lines 199-206`). When computing similarity with the newly generated *anchor token*, the anchor token at each selection step is represented by a **single $1 \times d_h$ hidden-state vector** (`Line 209`). Computing its distances to all stored tool embeddings therefore reduces to inexpensive vector operations.
>
> In practice, the embedding lookup and distance computations take **microseconds to low milliseconds**, whereas a single LLM forward pass typically takes **tens to hundreds of milliseconds**. We further conduct a time-overhead analysis showing that tool-selection operations contribute to only **≈0.8%** of the total end-to-end inference time. Thus, the embedding-based tool-selection step is negligible relative to the dominant LLM inference cost and will not become a runtime bottleneck even as the toolset scales further.
>
> **(iii) Generalizability of AutoTool across various downstream tasks.**
>
> In the experiment section, we demonstrate that our curated toolset, paired with AutoTool’s automatic selection **has supported a diverse set of tasks spanning math and science reasoning, search, and multimodal understanding** with strong performance. This goes beyond prior works [1,2,3,5], where agents leverage tools only within a single specialized domain or task. These results indicate that AutoTool is generalizable to cover a wide range of domains at the current scale.
>
> **(iv) Additional extensions to further scale AutoTool with more tools.**
>
> Our current embedding-based selection has **O(N)** linear complexity, which is relatively lightweight in practice. Following the reviewer's suggestion, we **outline several potential integrations** that could further improve the efficiency of AutoTool’s tool-selection process.
> - **Frequent-Tool Buffer:** Tools can be dynamically re-ranked based on usage statistics to form a high-frequency **priority buffer**. AutoTool can then compare the anchor vector against this smaller buffer first, falling back to the full set only when necessary.
> - **Hierarchical Tool Search:** A hierarchical search structure (e.g., clustering-based retrieval or approximate nearest-neighbor libraries such as FAISS/ScaNN) can reduce lookup from **O(N)** to **sublinear** time while scaling to millions of tools. Since Eq. 4 operates in the embedding space, replacing exact search with the hierarchical search is straightforward and requires no change to the policy model.
>
> **References:**
>
> [1] ReTool: Reinforcement Learning for Strategic Tool Use in LLMs.\
> [2] Search-r1: Training llms to reason and leverage search engines with reinforcement learning.\
> [3] Openthinkimg: Learning to think with images via visual tool reinforcement learning.\
> [4] Agentic Reasoning and Tool Integration for LLMs via Reinforcement Learning.\
> [5] ToolRL: Reward is All Tool Learning Needs.
>
> ---
> ## Happy to have further discussion!
> **Thank you again for the thoughtful review. We hope our responses address your concerns and are happy to discuss any further questions!**

---

> > ### Comment · Reviewer_NAkE · 2025-11-22
> > **thanks for the reply**
> >
> > Thank authors for the replies.
> > Would it be possible to compare AutoTool with Toolformer, and ToolGen ?

---

> > > ### Author Response · Authors · 2025-11-22
> > > **Follow-up to Reviewer NAkE's Reply**
> > >
> > > **Dear Reviewer NAkE,**
> > >
> > > **Thank you for your prompt reply and the insightful question! We will provide the detailed response to your follow-up question below.**
> > >
> > > ---
> > > > ## ***"Would it be possible to compare AutoTool with Toolformer, and ToolGen?"***
> > >
> > > ## RoadMap
> > > We first (i) highlight the **key methodological differences and novelty contributions of AutoTool**, and then (ii) present new **comprehensive experiments** demonstrating AutoTool’s strong effectiveness and generalizability compared with Toolformer and ToolGen.
> > > ## Response
> > >
> > > ### **(i) Methodology and Objective Comparison**
> > >
> > >
> > > We first highlight the **core differences and contributions of AutoTool** compared with the two related works in the following table.
> > >
> > >
> > >
> > >
> > >
> > > | Method | Core Objective | Toolset Assumption | Tool Representation | Tool Type | Generalization to Unseen Tools |
> > > |-|-|-|-|-|-|
> > > | Toolformer [1] | Improve LM via API call insertion | Small, fixed tool list | API templates hard-coded in text | only APIs | Not supported|
> > > | ToolGen [2] | Build tool-use agents over static APIs | Static Repository | Tool tokens hard-coded to vocab |only APIs| Limited (must re-tokenize & re-pretrain to add tools) |
> > > | **AutoTool (Ours)** | **Learn dynamic, step-wise tool selection for reasoning agents** | **Evolving toolset with unseen tools at inference** | **Continuous and generalizable embeddings of tool name + description** | **Both online and offline tools (covering code, math, muldimodal, etc.)** | **Yes** |
> > >
> > > **[Key Summary]**
> > >
> > > **1. Relation to Toolformer and ToolGen.** Toolformer and ToolGen are complementary to our work but **operate under different assumptions**:
> > > - Toolformer augments a pretrained LM with self-supervised API calls for a small, fixed set of tools, without explicit supervision for multi-step tool selection or evolving toolsets.
> > > - ToolGen virtualizes each tool as a dedicated token and trains the LLM to generate these tokens from queries and trajectories over a static repository APIs, mainly focusing on retrieval metrics and experiments.
> > >
> > > **2. Contribution of AutoTool.** AutoTool is explicitly designed for dynamic tool selection **under evolving and unseen toolsets.**
> > > - We curate a **dynamic tool–task ecosystem** with 1,346 tools, where 65.8% are held out and only appear at inference. We further inject explicit tool-selection rationales to enable LLM agents to **perform auto-regressive tool selection during generation**.
> > > - We introduce a dual-phase RL + Pl-ranking objective that aligns the **policy’s embedding-anchored selection** with **reward-consistent preferences** over sequences of tools.
> > >
> > > Our method design enables AutoTool to **generalize to entirely new tools at inference, which Toolformer and ToolGen cannot natively handle without retraining or re-virtualizing the toolset.**
> > >
> > >
> > > ### **(ii) Experimental Comparison**
> > >
> > > To better demonstrate the effectiveness and generalizability of AutoTool, we conduct new experiments to compare with Toolformer and ToolGen.
> > >
> > > **[Experiment Setting]** We compare three methods on both **in-domin tasks (AIME 2024, GPQA-Diomond)** and **out-of-domain tasks (LiveCodeBench, MedQA).** We use AutoTool (Qwen3-8B) as our evaluation model.
> > >
> > > **[Evaluation Results]**
> > > |Method|AIME24|GPQA-Diamond|LiveCodeBench|MedQA|
> > > |-|-|-|-|-|
> > > |Toolformer|6.7|23.2|10.8|23.7|
> > > |ToolGen|13.3|38.4|46.1|56.0|
> > > |**AutoTool**|**68.8**|**73.7**|**77.9**|**84.3**|
> > >
> > > **Key Findings.** From the experiemnts, we observe that:
> > >
> > > - **AutoTool consistently outperforms Toolformer and ToolGen (by 60.1%, 37.7%) across all benchmarks**, confirming its stronger capability in accurate, step-wise tool selection.
> > > - **AutoTool remains highly effective on out-of-domain tasks (e.g., LiveCodeBench, MedQA)**, demonstrating robust generalization and reliability under evolving, unseen tool environments.
> > >
> > >
> > > **References:**\
> > > [1] Toolformer: Language Models Can Teach Themselves to Use Tools.\
> > > [2] Toolgen: Unified tool retrieval and calling via generation.
> > >
> > > ---
> > >
> > > **We sincerely thank the reviewer for this insightful question and ensure to revise our related work and experiment section to include a detailed discussion of these two works and cite them appropriately in our revision.**

---

> ### Author Response · Authors · 2025-11-27
> **Looking Forward to Your Insightful Feedback!**
>
> Dear Reviewer NAkE,
>
> We sincerely thank you for your thoughtful and constructive review, which has greatly helped us improve our manuscript! In response to your original review and additional follow-up questions, we have conducted detailed experiments and in-depth analyses of our method.
>
> As the discussion period draws to a close, we would appreciate your feedback on whether our responses have addressed your concerns. We are fully committed to addressing any remaining questions you may have. Thank you very much for your time and consideration of our research!
>
> Warm regards,
>
> Authors of AutoTool

---

### Author Response · Authors · 2025-12-03
**Summary of Reviews, Paper Revision, and Rebuttal. Thanks for Your Efforts! (AOE 12/02/2025)**

Dear PCs, SACs, ACs, and Reviewers,

Thank you very much for your guidance and valuable contributions to our work. We also sincerely appreciate the reviewers’ thoughtful and constructive feedback, and we are encouraged by the positive assessments following our rebuttal.

In response to the insightful comments, **we have conducted substantial new experiments and in-depth analyses, and accordingly revised the manuscript ([click here](https://openreview.net/pdf?id=52c4trAbmd))** to improve both clarity and technical completeness. To assist the AC and help reduce their review workload, we briefly summarize the key points raised by the reviewers and our corresponding updates below.

---
## **Strength and Contributions**

**1. Significance of the problem setting (Reviewer NAkE, Rg4C)**: Reviewers agreed that addressing dynamic tool selection under evolving toolsets tackles a critical real-world limitation of fixed-inventory LLM agents.

**2. Well-designed dual-phase optimization pipeline (Reviewer NAkE, Rg4C)**: The dual-phase optimization framework of AutoTool was recognized as a well-motivated and theoretically sound training strategy.

**3. Large-scale dataset with tool-selection supervision (Reviewer NAkE, Rg4C)**: The 200k-trajectory dataset with explicit tool-selection rationales across 1,000+ tools and 100+ tasks was acknowledged as a substantial and valuable resource.

**4. Comprehensive and strong theoretical and empirical results (Reviewer HQaG, Rg4C)**: Reviewers highlighted the comprehensive evaluations and consistent performance gains over strong reasoning and tool-integration baselines across diverse benchmarks.

---
## **How We Addressed the Reviewers’ Concerns**

**1. Addressing Generalization to Unseen Tools via Dataset Curation and Method Design (Reviewers HQaG, Rg4C)**\
**Quick Link:** [Reply to HQaG](https://openreview.net/forum?id=52c4trAbmd&noteId=AQF9lGg9T0), [Reply to Rg4C](https://openreview.net/forum?id=52c4trAbmd&noteId=QRi6BF3Q8C)
- We clarified the **training–inference tool split** in our dataset with detailed statistics.
- We added **new tool-usage analyses** demonstrating that AutoTool selects a majority of unseen tools during successful inference across multiple datasets.
- Furthermore, we conducted **fully unseen-tool evaluations** on four benchmarks, where all inference-time tools are held out from training.

Together, these results provide direct and strong evidence of **AutoTool’s robust generalization to unseen and evolving toolsets**.

**2. Validating the Relationship between Tool Selection and Final Accuracy (Reviewer Rg4C)**\
**Quick Link:** [Reply to Rg4C](https://openreview.net/forum?id=52c4trAbmd&noteId=DxctzcF9rJ)

- We added **a new tool hit-rate analysis** comparing selected tools against oracle tools across multiple benchmarks.
- We demonstrated **a strong positive correlation** between tool-selection accuracy and final task performance.

**3. Scalability of AutoTool & Comparison to Prior Tool Methods (Reviewer NaKE)**\
**Quick Link:** [Reply to NaKE](https://openreview.net/forum?id=52c4trAbmd&noteId=e9w40nqeQw)
- We added **new scalability analysis** showing tool selection adds only ~0.8% end-to-end inference overhead at 1,000+ tools.
- We provided **detailed explanation and related-work comparisons** with related works such as ToolLLM, RestGPT, and HuggingGPT.

---
## **Reviewer's Engagement Summary During Rebuttal**
We sincerely thank all reviewers for their positive feedback and active engagement throughout the discussion period. As a summary:

**1. Reviewer NAkE** offered **a supportive and positive assessment with a score of 6** from the start.

**2. Reviewer Rg4C** confirmed that our new comprehensive analyses have **"addressed or clarified the reviewer's concerns."** and the reviewer **updated the overall rating to 6** on Nov. 24.

**3. Reviewer HQaG** has not yet had the opportunity to engage in the discussion. Nevertheless, the reviewer **assigned positive sub-scores (all 3: good) for Soundness, Presentation, and Contribution**, and **explicitly indicated a strong willingness to raise the scores if given detailed analyses on the generalization.**

As Reviewer HQaG’s concerns on generalization closely align with W.1 raised by Reviewer Rg4C, and we have fully addressed this issue through **comprehensive new analyses and unseen-tool evaluations**, we are confident that our detailed rebuttal and revised submission can resolve Reviewer HQaG’s concerns as well.

---
Once again, we sincerely thank the reviewers, AC, SAC, and PC for their careful reading and constructive feedback. **Your insightful comments have substantially strengthened both the technical rigor and the clarity of our paper!**

Warm Regards,

The Authors of AutoTool

---

### Meta-Review · Area_Chair_46ej · 2026-01-07

**Summary:**

The proposed framework, AutoTool, aims to solve a significant bottleneck in current LLM agent research: the reliance on fixed toolsets. Real-world applications often involve "evolving" toolsets where new APIs are added frequently. AutoTool uses a dual-phase optimization strategy and an embedding-anchored selection mechanism to generalize to tools it has never seen during training.

The review process saw a positive trend, with most reviewers acknowledging the significance of the problem and the scale of the authors' dataset (200k trajectories).

**Reviewer Concerns:**

The review process saw a positive trend, with most reviewers acknowledging the significance of the problem and the scale of the authors' dataset; however, the reviewers were also concerned about

Generalization to Unseen Tools (HQaG, Rg4C): addressed
Baselines (NAkE): The authors added direct comparisons to Toolformer and ToolGen, showing that AutoTool significantly outperforms them (addressed)

Theoretical vs. Empirical Scaling: While the 1,000-tool scale is well-tested, the transition to "millions of tools" remains a theoretical discussion

**Reviewer Scores:**

Rg4C and HQaG might consider raising their score.

---

### Decision · Program_Chairs · 2026-01-26

Reject